# Rapid Task-Solving in Novel Environments

**Samuel Ritter**\*, **Ryan Faulkner**\*, **Laurent Sartran**, **Adam Santoro**,
**Matthew Botvinick**, **David Raposo**

DeepMind
London, UK

{ritters, rfaulk, lsartran, adamsantoro,
botvinick, draposo}@google.com

## Abstract

We propose the challenge of rapid task-solving in novel environments (RTS), wherein an agent must solve a series of tasks as rapidly as possible in an unfamiliar environment. An effective RTS agent must balance between exploring the unfamiliar environment and solving its current task, all while building a model of the new environment over which it can plan when faced with later tasks. While modern deep RL agents exhibit some of these abilities in isolation, none are suitable for the full RTS challenge. To enable progress toward RTS, we introduce two challenge domains: (1) a minimal RTS challenge called the Memory&Planning Game and (2) One-Shot StreetLearn Navigation, which introduces scale and complexity from real-world data. We demonstrate that state-of-the-art deep RL agents fail at RTS in both domains, and that this failure is due to an inability to plan over gathered knowledge. We develop Episodic Planning Networks (EPNs) and show that deep-RL agents with EPNs excel at RTS, outperforming the nearest baseline by factors of 2-3 and learning to navigate held-out StreetLearn maps within a single episode. We show that EPNs learn to execute a value iteration-like planning algorithm and that they generalize to situations beyond their training experience.

## 1 Introduction

An ideal AI system would be useful immediately upon deployment in a new environment, and would become more useful as it gained experience there. Consider for example a household robot deployed in a new home. Ideally, the new owner could turn the robot on and ask it to get started, say, by cleaning the bathroom. The robot would use general knowledge about household layouts to find the bathroom and cleaning supplies. As it carried out this task, it would gather information for use in later tasks, noting for example where the clothes hampers are in the rooms it passes. When faced with its next task, say, doing the laundry, it would use its newfound knowledge of the hamper locations to efficiently collect the laundry. Humans make this kind of *rapid task-solving in novel environments* (RTS) look easy (Lake et al., 2017), but as yet it remains an aspiration for AI.

Prominent deep RL systems display some of the key abilities required, namely, exploration and planning. But, they need many episodes over which to explore (Ecoffet et al., 2019; Badia et al., 2020) and to learn models for planning (Schrittwieser et al., 2019). This is in part because they treat each new environment in isolation, relying on generic exploration and planning algorithms. We propose to overcome this limitation by treating RTS as a meta-reinforcement learning (RL) problem, where agents *learn exploration policies and planning algorithms* from a distribution over RTS challenges. Our contributions are to:

1. Develop two domains for studying meta-learned RTS: the minimal and interpretable Memory&Planning Game and the scaled-up One-Shot StreetLearn.

2. Show that previous meta-RL agents fail at RTS because of limitations in their ability to plan using recently gathered information.

---

\*Equal contribution.

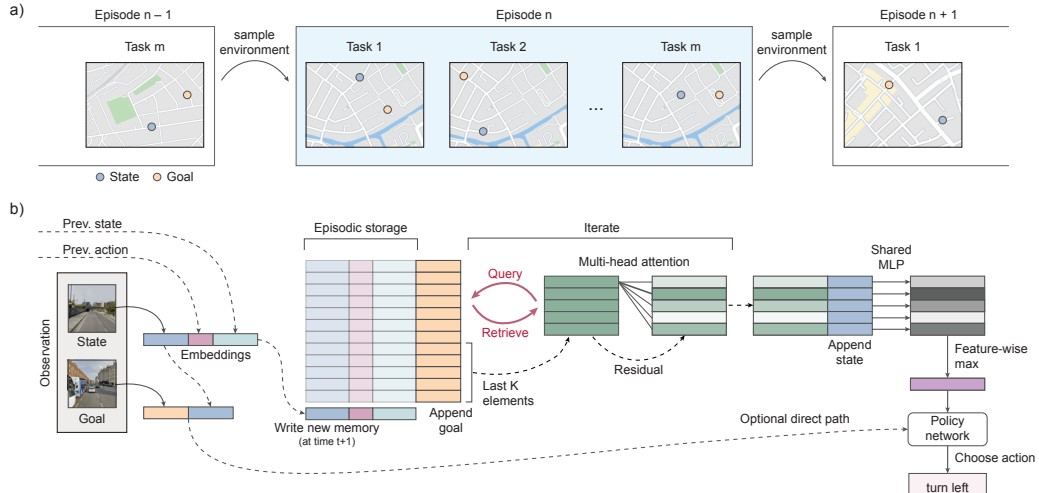

Figure 1: (a) *Rapid Task Solving in Novel Environments* (RTS) setup. A new environment is sampled in every episode. Each episode consists of a sequence of tasks which are defined by sampling a new goal state and a new initial state. The agent has a fixed number of steps per episode to complete as many tasks as possible. (b) *Episodic Planning Network* (EPN) architecture. The EPN uses multiple iterations of a single shared self-attention function over memories retrieved from an episodic storage.

3. Design a new architecture – the Episodic Planning Network (EPN) – that overcomes this limitation, widely outperforming prior agents across both domains.

4. Show that EPNs learn exploration and planning algorithms that generalize to larger problems than those seen in training.

5. Demonstrate that EPNs learn a value iteration-like planning algorithm that iteratively propagates information about state-state connectivity outward from the goal state.

## 2   PROBLEM FORMULATION

Our objective is to build agents that can maximize reward over a sequence of tasks in a novel environment. Our basic approach is to have agents *learn* to do this through exposure to distributions over multi-task environments. To define such a distribution, we first formalize the notion of an environment $e$ as a 4-tuple $(S, A, P_a, \mathcal{R})$ consisting of states, actions, a state-action transition function, and a *distribution over* reward functions. We then define the notion of a task $t$ in environment $e$ as a Markov decision process (MDP) $(S, A, P_a, r)$ that results from sampling a reward function $r$ from $\mathcal{R}$.

We can now define a framework for learning to solve tasks in novel environments as a simple generalization of the popular meta-RL framework (Wang et al., 2017; Duan et al., 2016). In meta-RL, the agent is trained on MDPs $(S, A, P_a, r)$ sampled from a task distribution $\mathcal{D}$. In the rapid task-solving in novel environments (RTS) paradigm, we instead sample problems by first sampling an environment $e$ from an environment distribution $E$, then sequentially sampling tasks, i.e. MDPs, from that environment's reward function distribution (see Figure 1).

An agent can be trained for RTS by maximizing the following objective:

$$\mathbb{E}_{e \sim E}\big[\mathbb{E}_{r \sim \mathcal{R}_e}[J_{e,r}(\theta)]\big],$$

where $J_{e,r}$ is the expected reward in environment $e$ with reward function $r$. When there is only one reward function per environment, the inner expectation disappears and we recover the usual meta-RL objective $\mathbb{E}_{e \sim E}[J_e(\theta)]$. RTS can be viewed as meta-RL with the added complication that the reward function changes within the inner learning loop.

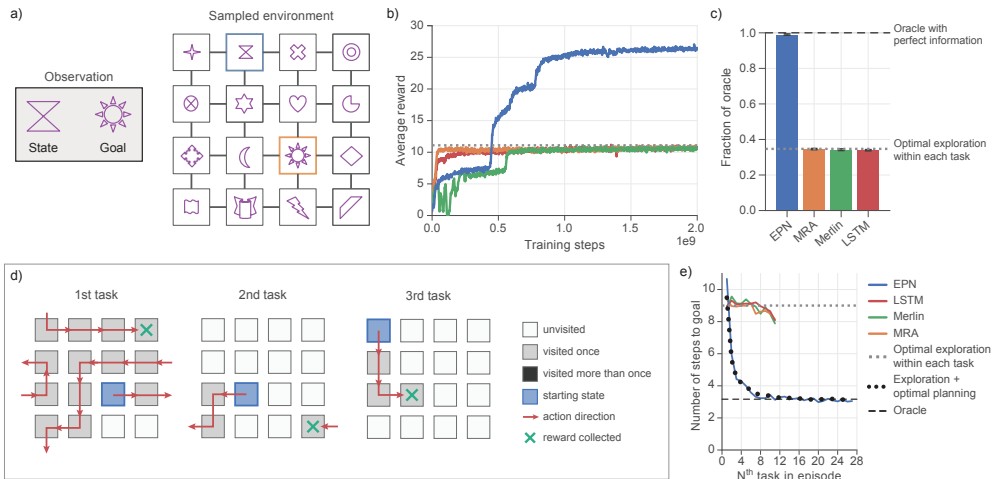

Figure 2: Memory&Planning Game. (a) Example $4 \times 4$ environment (not observable by the agent) and state-goal observation. (b) Training curves. Performance measured by the average reward per episode, which corresponds to the average number of tasks completed within a 100-step episode (showing the best runs from a large hyper-parameter sweep for each model). (c) Performance measured in the last third of the episodes (post-training), relative to an oracle with perfect information that takes the shortest path to the goal. (d) Example trajectory of a trained EPN agent in the first three tasks of an episode. In the first task, the agent explores optimally without repeating states. In the subsequent tasks, the agent takes the shortest path to the goal. (e) Number of steps taken by an agent to reach the $n^{th}$ goal in an episode.

While meta-RL formalizes the problem of learning to learn, RTS formalizes both the problems of learning to learn and learning to plan. To maximize reward while the reward function is constantly changing in non-trivial novel environments, agents must learn to (1) efficiently explore and effectively encode the information discovered during exploration (i.e., *learn*) and (2) use that encoded knowledge to select actions by predicting trajectories it has never experienced (i.e., *plan*[1]).

## 3   THE LIMITATIONS OF PRIOR AGENTS

To test whether past deep-RL agents can explore and plan in novel environments, we introduce a minimal problem that isolates the challenges of (1) exploring and remembering the dynamics of novel environments and (2) planning over those memories. The problem we propose is a simple variation of the well-known Memory Game[2], wherein players must remember the locations of cards in a grid. The variation we propose, which we call the Memory&Planning Game, extends the challenge to require planning as well as remembering (see Figure 2).

In the Memory&Planning Game, the agent occupies an environment consisting of a grid of symbols ($4 \times 4$). The observation consists of two symbols – one which corresponds to the agent's current location, and another that corresponds to the "goal" the agent is tasked with navigating to. The agent can not see its relative location with respect to other symbols in the grid. At each step the agent selects one of 5 possible actions: *move left*, *move right*, *move up*, *move down*, and *collect*. If the agent chooses the "collect" action when its current location symbol matches the goal symbol, a reward of $1$ is received. Otherwise, the agent receives a reward of $0$. At the beginning of each episode, a new set of symbols is sampled, effectively inducing a new transition function. The agent is allowed a fixed number of steps (100) per episode to "collect" as many goals as possible. Each time the agent collects a goal – which corresponds to completing a task –, a new goal is sampled in and the transition function stays fixed.

---

[1]Following Sutton & Barto (1998), we refer to the ability to choose actions based on predictions of not-yet-experienced events as "planning".

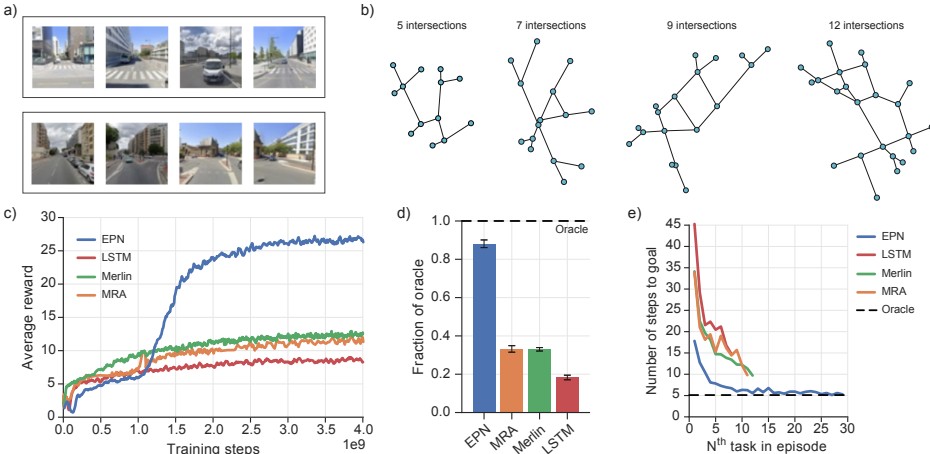

Figure 3: One-Shot StreetLearn. (a) Four example states from two randomly sampled neighborhoods. (b) Example connectivity graphs. (c) Evaluation performance measured on neighborhoods of a held-out city throughout the course of training (showing the best run from a large hyper-parameter sweep for each model). (d) Performance in last third of episode relative to an oracle with perfect information. (e) Number of steps taken by an agent to reach the $n^{th}$ goal in an episode.

A successful agent will (1) efficiently explore the grid to discover the current set of symbols and their connectivity and (2) plan shortest paths to goal symbol if it has seen (or can infer) all of the transitions it needs to connect its current location and the current goal location. This setup supplies a minimal case of the RTS problem: at the beginning of each episode the agent faces a new transition structure $(S, A, P_a)$ and must solve the current task while finding information that will be useful for solving future tasks on that same transition structure. In subsequent tasks, the agent must use its stored knowledge of the current grid's connectivity to plan shortest paths.

In this symbolic domain, we find evidence that previous agents learn to explore but not to plan. Specifically, they match the within-trial planning optimum; that is, a strategy that explores optimally within each task, but forgets everything about the current environment when the task ends (Figure 2). We hypothesize that this failure is the result of the limited expressiveness of past architectures' mechanisms for processing retrieved memories. We designed Episodic Planning Networks to overcome this limitation.

## 4 EPISODIC PLANNING NETWORKS

Past episodic memory agents used their memory stores by querying the memories, summing the retrieved slots, then projecting the result through a multi-layered perceptron (MLP) (Fortunato et al., 2019; Wayne et al., 2018). We hypothesize that these agents fail to plan because the weighted sum of retrieved slots is not sufficiently *representationally* expressive, and the MLP is not, on its own, sufficiently *computationally* expressive to support planning in non-trivial environments. To test this hypothesis, we replace the weighted sum and MLP with an an iterative self-attention-based architecture designed to support implicit planning.

The architecture, which we call the Episodic Planning Network (EPN, see Figure 1b) consists of three main components: (1) an episodic memory which reflects the agent's experience in the episode so far; (2) an iterative self-attention module that operates over memories, followed by a pooling operation (we will refer to this component as the *planner*); and (3) a policy network.

**Episodic memory –** At each timestep, we start by appending the embedding of the current goal to each slot in the episodic memory. The result is passed as input to the planner. We then add a new

---

[2]In a striking parallel, the Memory Game played a key role in the development of memory abilities in early episodic memory deep-RL agents (Wayne et al., 2018).

memory to the episodic memory, which represents that timestep's transition: it is the concatenation of embeddings of the current observation, the previous action, and the previous observation.

**Planner –** A self-attention-based update function is applied over the memories to produce a processed representation of the agent's past experience in the episode and reflecting the agent's belief about the environment. This self-attention operation is iterated some number of times, sharing parameters across iterations. The current state is appended to each element resulting from the self-attention, each of which is then passed through a shared MLP. The resulting vectors are aggregated by a feature-wise max operation.

Note that the self-attention (SA) does not have access to the current state until the very end – it does all of its updates with access only to the goal state and episodic memories. This design reflects the intuition that the SA function might learn to compute something like a value map, which represents the distance to goal from all states in memory. If the final SA state does in fact come to represent such a value map, then a simple MLP should be sufficient to compute the action that, from the current state, leads to the nearest/highest value state with respect to the goal. We find in Section 5, Figure 4 evidence that the final SA state comes in fact to resemble an iteratively improved value map.

The specific self-attention update function we used was:

$$B_{i+1} = f(B_i + \phi(B_i))$$
$$\phi(x) = \text{MHA}(\text{LayerNorm}(x))$$

where MHA is the multi-head dot-product attention described by Vaswani et al. (2017). Layer normalization was applied to the inputs of each iteration, and residual connections were added around the multi-head attention operation.[3] In our experiments, $f$ was a ReLU followed by a 2-layer MLP shared across rows.

**Policy network –** A 2-layer multi-layer perceptron (MLP) was applied to the output of the planner to produce policy logits and a baseline.

When using pixel inputs, embeddings of the observations were produced by a 3-layer convolutional neural network. We trained all agents to optimize an actor-critic objective function using IMPALA, a framework for distributed RL training (Espeholt et al., 2018). For further details, please refer to the Supplemental Material.

Our results show that the EPN succeeds in both exploration and planning in the Memory&Planning Game. It matches the shortest-path optimum after only a few tasks in a novel environment (Figure 2c), and its exploration and planning performance closely matches the performance of a hand-coded agent which combines a strong exploration strategy – avoiding visiting the same state twice – with an optimal planning policy (Figure 2d,e). For these results and the ones we will present in the following section, all of the episodic memories attend to all of the others. This update function scales quadratically with the number of memories (and the number of steps per episode). In the Supplemental Material we present results showing that we recover 92% of the performance using a variant of the update function that scales linearly with the size of the memory (see Figure 5 in the Suppl. Material).

## 5   ONE-SHOT STREETLEARN

We now test whether our agent can extend its success in the Memory&Planning Game to a domain with high-dimensional pixel-based state observations, varied real-world state-connectivity, longer planning depths, and larger time scales. We introduce the One-Shot StreetLearn domain (see Figure 1a), wherein environments are sampled as neighborhoods from the StreetLearn dataset of Google StreetView images and their connectivity (Mirowski et al., 2019). Tasks are then sampled by selecting a position and orientation that the agent must navigate to from its current location.

In past work with StreetLearn, agents were faced with a single large map (e.g. a city), in which they could learn to navigate over billions of experiences (Mirowski et al., 2018). Here, we partition the

---

[3]Applying the layer normalization to the output of the multi-head attention, or removing the layer normalization altogether, did not produce good results in our experiments.

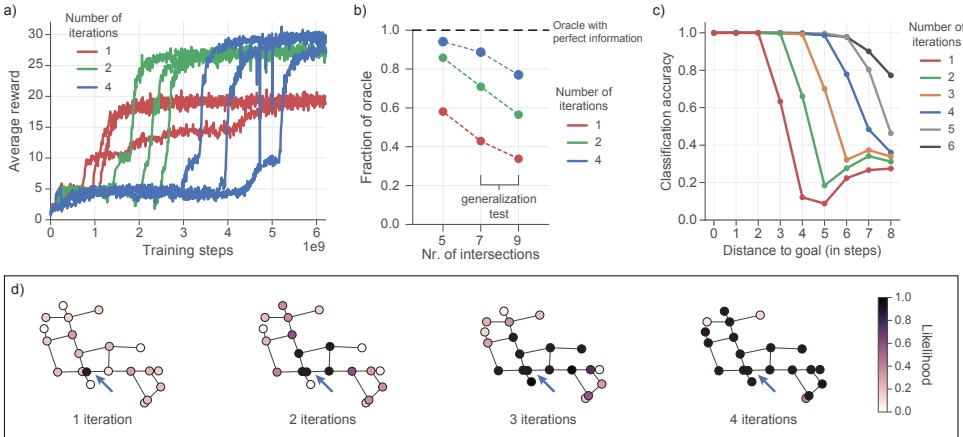

Figure 4: Iteration analysis and generalization. (a) Performance of an EPN agent on held-out neighborhoods with 5 intersections using planners with 1, 2 and 4 self-attention iterations (showing 3 runs for each condition). (b) Performance on new neighborhoods that were larger (7 and 9 intersections) than the ones used during training (5 intersections). (c) Distance-to-goal decoding accuracy from the output of the planner after 1 to 6 iterations. (d) The ability of EPN activations to predict distance to goal expands out from the goal state (blue arrow) as the number of self-attention iterations increases. See section 6.2 for details.

StreetLearn data into a many different maps (e.g. neighborhoods of different cities), so that agents can be trained to rapidly solve navigation tasks in new maps. This approach is analogous to the partitioning of the ImageNet data into many one-shot classification tasks that spurred advances in one-shot learning (Vinyals et al., 2016). In this case, rather than learning to classify in one shot, agents should learn to *plan* in one shot, that is, to plan after one or fewer observations of each transition in the environment.

In One-Shot StreetLearn, the agent's observations consist of an image representing the current state and an image representing the goal state (Figure 1b). The agent receives no other information from the environment. The available actions are *turn right*, which orients the agent clockwise toward the next available direction of motion from its current node; *turn left*, which does the same in the other direction; and *move forward*, which moves the agent along the direction it is facing to the next available node. In each episode, we sample a new neighborhood with 5 intersections from one of 12 cities. To reduce the exploration problem while keeping the planning difficulty constant, we removed all the locations between intersections that corresponded to degree-2 nodes. This resulted in graphs with a median of 12 nodes (see example connectivity graph in Figure 3b, leftmost panel). Every time the agent reaches a goal, a new starting- and goal-state pair is sampled, initiating a new task, until the fixed episode step limit is reached (200 steps). Neighborhoods with 5 intersections allow us to sample from approximately 380 possible tasks (i.e. starting- and goal-state pairs) ensuring a small chance of repeating any single task. A step that takes the agent to the goal state results in a reward of 1. Any other step results in a reward of 0.

One-Shot StreetLearn has several important characteristics for developing novel-environment planning agents. First, we can generate many highly varied environments for training by sampling neighborhoods from potentially any place in the world. This allows us to generate virtually unlimited maps with real-world connectivity structure for agents to learn to take advantage of. Second, the agent's observations are rich, high-dimensional visual inputs that simulate a person's experience when navigating. These inputs can in principle be used by an agent to infer some of the structure of a new environment. Third, we can scale the planning depth arbitrarily by changing the size of the sampled neighborhoods – a useful feature for iteratively developing increasing capable planning systems. Fourth, the time-scale of planning can be varied independently of the planning depth by varying the number of nodes between intersections. This is a useful feature for developing temporally abstract (i.e. jumpy) planning abilities (see Section 7 for discussion).

**EPNs Outperform All Baselines –** We trained EPN-based agents and previous deep-RL agents on One-Shot StreetLearn neighborhoods with 5 intersections. Our agent significantly outperformed the baselines, successfully reaching 28.7 goals per episode (averaged over 100 consecutive episodes, Figure 3c). It is important to note that this performance was measured on neighborhoods of a held-out city, which the agent was visiting for the first time. Baseline agents with an episodic memory system (Merlin and MRA) did not exceed 14.5 goals per episode, performing better than ones without (LSTM) that were only able to reach 10.0.

**EPNs optimally navigate new maps after 10-15 tasks –** Plotting the average number of steps taken by the agent to complete the $n^{th}$ task of an episode reveals that the agents needs fewer and fewer steps to complete a new task, matching the minimum number of steps required to reach a new goal after 10–15 tasks (Figure 3e). In the last third of every episode, EPNs achieve on average $88\% \pm 2\%$ of the performance of an oracle with perfect information that systematically takes the shortest-path to the goals (Figure 3d). We chose the last third of the episodes for this comparison as a way to exclude steps used for exploration and thus have a better estimate of the agent's planning capabilities.

**EPNs Generalize to Larger Maps –** We trained EPN agents on neighborhoods with 5 intersections and evaluated them (with gradient learning turned off) on larger neigbhoroods with 7 and 9 intersections. We found that the trained agents achieved strong performance relative to the oracle, suffering only small marginal drops in performance even on maps nearly double the size of those seen during training (Figure 4b, in blue).

**More "thinking time" improves performance –** A desireable property for a planning algorithm would be for performance to increase as "thinking time" increases, allowing the user to obtain better performance simply by allowing more computation time. We test whether EPNs have this property by evaluating the effect of the number of self-attention iterations (i.e., "thinking time") on evaluation performance. We observed a systematic performance boost with each additional iteration used in training[4] (see Figure 4a). When evaluated on neighborhoods with 5 intersections (same neighborhood size as the ones used during training) the EPN agent's performance approaches saturation with 2 or more iterations, as indicated by the fraction-of-oracle performance measure in Figure 4b (leftmost points). When we increase the size of the neighborhoods (7 and 9 intersections) the performance difference between 2 and 4 iterations becomes more pronounced, as indicated by the divergence between the blue and green dashed lines in Figure 4b.

**Trained EPNs recursively refine a value-like map –** To demonstrate this we froze the weights of a trained EPN agent and replaced its policy network with an MLP decoder of the same size. We then trained the decoder (with a stop gradient to the planner) to output the distance from a random state to a random goal, while providing the planner with memories containing all the transitions of a randomly sampled environment. We repeated this experiment with six decoders that received inputs from the planner after a fixed number of iterations, from 1 to 6.[5]

The training loss of the different decoders revealed a steady increase in the ability to compute distance to goal as we increase the number of self-attention iterations from 1 to 6 (see Figure 6a in the Suppl. Material). This gain is a direct consequence of improved decoding for longer distances, as made evident by the gradual rightward shift in the drop-off of the classification accuracy plotted against distance to goal (Figure 4c). This holds true even when we increase the number of iterations beyond the number of iterations used during the training of the planner. Altogether, this suggests that the planner is able to "look" further ahead when given more iterations.

We can visualize this result by selecting a single evaluation environment (in this case, a larger neighborhood with 12 intersections) with a fixed goal location, and measuring the likelihood resulting from all possible states in that environment. This manipulation reveals a spatial pattern in the ability to compute distance to goal, spreading radially from the goal location as we increase the number of self-attention iterations (see Figure 4d and Figure 6b in the Suppl. Material).

**Ablations –** Besides providing a strong baseline, MERLIN/MRA also represent a key ablation for understanding the causes of EPNs' performance: they have everything the EPN agent has *except the planning module*. LSTMs represent the next logical ablation: they lack both episodic memory and

---

[4]The experiments described here were restricted to one-hot inputs for quicker turnaround.

[5]Note that while the decoders were trained in a supervised setting using 1 to 6 iterations, the weights of the planner were trained once in the RL setting using 4 iterations.

the planning module. The performance across these three types of architectures shows that episodic memory alone helps very slightly, but is not enough to produce planning behavior; for that, the self-attention module is necessary.

Another important ablation study is the progressive removal of the later planning iterations (Figure 4a). From this experiment we learned that one step of self-attention isn't enough: iterating the planning module is necessary for full performance.

## 6 RELATED WORK

Deriving repurposable environment knowledge to solve arbitrary tasks has been a long-standing goal in the field of RL (Foster & Dayan, 2002; Dayan, 1993) toward which progress has recently been made (Sutton et al., 2011; Schaul et al., 2015; Borsa et al., 2018). While quick to accommodate new tasks, these approaches require large amounts of time and experience to learn about new environments. This can be attributed to their reliance on gradient descent for encoding knowledge and their inability to bring prior knowledge to new environments (Botvinick et al., 2019). Our approach overcomes these problems by recording information about new environments in activations instead of weights and learning prior knowledge in the form of exploration and planning policies.

The possibility of learning to explore with recurrent neural networks was demonstrated by Wang et al. (2017) and Duan et al. (2016) in the context of bandit problems. Wang et al. (2017) further demonstrated that an LSTM could learn to display the behavioral hallmarks of model-based control – in other words, *learn to plan* – in a minimal task widely used for studying planning in humans Daw et al. (2011). The effectiveness of learned, implicit planning was demonstrated further in fully-observed spatial environments using grid-structured memory and convolutional update functions (Tamar et al., 2016; Lee et al., 2018b; Guez et al., 2019). Gupta et al. (2017) extended this approach to partially observed environments with ground-truth ego-motion by training agents to emulate an expert that computes shortest paths. Our present work can be seen as extending learned planning by (1) solving partially-observed environments by learning to gather the information needed for planning, and (2) providing an architecture appropriate for a broader range of (e.g. non-spatial) planning problems.

Model-based reinforcement learning (MBRL) has long been concerned with building agents capable of constructing and planning over environment models (for review see Moerland et al., (2020)). However, we are not aware of any MBRL agents that learn a deployable model *within a single episode of experience*, as is required for the RTS setting.

Learning to plan can be seen as a specific case of learning to solve combinatorial optimization problems, a notion that has been taken up by recent work (for review see Bengio et al., 2018). Especially relevant to our work is Kool et al (2018), who show that transformers can learn to solve large combinatorial optimization problems, comparing favorably with more resource-intensive industrial solvers. This result suggests that in higher-depth novel-environment planning problems than we consider in our current experiments, the transformer-based architecture may continue to be effective.

Savinov et al. (2018) develop an alternative to end-to-end learned planning that learns a distance metric over observations for use with a classical planning algorithm. Instead of learning to explore, this system relies on expert trajectories and random walks. It's worth noting that hand-written planning algorithms do not provide the benefits of domain-adapted planning demonstrated by Kool et al. (2018), and that design modifications would be needed to extend this approach to tasks requiring *abstract planning* - e.g. jumpy planning and planning over belief states - whereas the end-to-end learning approach can be applied to such problems out-of-the-box.

Recent work has shown episodic memory to be effective in extending the capabilities of deep-RL agents to memory intensive tasks (Oh et al., 2016; Wayne et al., 2018; Ritter et al., 2018b; Fortunato et al., 2019). We chose episodic memory because of the following desirable properties. First, because the episodic store is non-parametric, it can grow arbitrarily with the complexity of the environment, and approximate k-nearest neighbors method make it possible to scale to massive episodic memories in practice, as in Pritzel et al. (2017). This means that memory fidelity need not decay with time. Second, episodic memory imposes no particular assumptions about the environment's stucture, making it a potentially appropriate choice for a variety of non-spatial applications such as

chemical synthesis Segler et al. (2018) and web navigation Gur et al. (2018), as well as abstract planning problems of interest in AI research, such as Gregor et al.'s (2019) Voxel environment tasks.

Our approach can be seen as implementing episodic model-based control (EMBC), a concept recently developed in cognitive neuroscience (Vikbladh et al., 2017; Ritter, 2019). While episodic control (Gershman & Daw, 2017; Lengyel & Dayan, 2008; Blundell et al., 2016) produces value estimates using memories of individual experiences in a model-free manner, EMBC uses episodic memories to inform a model that predicts the outcomes of actions. Ritter et al. (2018a) showed that a deep-RL agent with episodic memory and a minimal learned planning module (an MLP) could learn to produce behavior consistent with EMBC (Vikbladh et al., 2017). Our current work can be seen as using iterated self-attention to scale EMBC to much larger implicit models than the MLPs of past work could support.

# 7 FUTURE WORK

We showed that deep-RL agents with EPNs can meta-learn to explore, build models on-the-fly, and plan over those models, enabling them to rapidly solve sequences of tasks in unfamiliar environments. This demonstration paves the way for important future work.

*Temporally abstract planning* (or, "jumpy" planning, Akilesh et al., 2019) may be essential for agents to succeed in temporally extended environments like the real world. our method makes no assumptions about the timescale of planning, unlike other prominent approaches to learned planning (Schrittwieser et al., 2019; Racanière et al., 2017). Training and evaluating EPNs on problems that benefit from jumpy planning, such as One-Shot StreetLearn with all intermediate nodes, may be enough to obtain strong temporal abstraction performance.

*Planning over belief states* may be essential for agents to succeed in dynamic partially-observed environments. Planning over belief states might be accomplished simply by storing belief states such as those developed by Gregor et al. (2019) and training on a problem distribution that requires planning over information stored in those states.

Humans are able to solve a seemingly *open-ended variety of tasks*, as highlighted for example by the Frostbite Challenge (Lake et al., 2017). The EPN architecture is in principle suitable for any task class wherein the reward function can be represented by an input vector, so future work may test EPNs in RTS problems with broader task distributions than those addressed in this work, e.g., by using generative grammars over tasks or by taking human input (Fu et al., 2019).

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

# A  APPENDIX

## A.1  ONE-SHOT STREETLEARN

**Dataset preparation** – The StreetLearn dataset Mirowski et al. (2019) provides a connectivity graph between nodes, and panoramic images (panos) for each node. As movement in the environment is limited to turning left, turning right, or moving forward to the next node, we only need from the panoramic images the frames corresponding to what an observer would see in a given location, when oriented towards a next possible location. This defines a mapping from oriented edge in the full StreetLearn graph to a single frame. We compute it ahead of time and store it in a data structure enabling efficient random accesses.

**Observation associated to an edge** – As our agent operates on a reduced graph containing only degree-2 nodes, we define the observation to an oriented edge A–B in the reduced graph to be the image associated to the first edge in the shortest path from A to B on the full graph.

**Movement** – When the agent moves forward, its new location becomes the neighbor it was oriented towards, and is reoriented towards the neighbor most compatible with its current orientation.

**Distributed learning setup** –

Each actor is assigned to a random city, out of 12 cities/regions in Europe: Amsterdam, Brussels, Dublin, Lisbon, Madrid, Moscow, Rome, Vienna, Warsaw, Paris North-East, Paris North-West and Paris South-West. A special *evaluator* actor, which does not send send trajectories to the learner, is assigned to a withheld region: Paris South-East.

**Sampling neighborhoods** – In the beginning of a new episode, we start by sampling a random location in the region assigned to the actor – i.e. a random node of the region's connectivity graph. This node will become the center-node of the new neighborhood and is added as the first node of the sampled graph. We then proceed to traverse the connectivity graph, breadth first, adding visited nodes to the sampled graph, until it contains the number of intersections required. An intersection is a node with degree greater than 2; the number of intersections is a parameter of environment that we can set for each experiment and it determines the difficulty (depth) of the planning problem. In our experiments we decided to remove all degree-2 nodes of the sampled graph. This manipulation allowed us to simplify the exploration problem substantially without reducing the planning difficulty.

## A.2  RL SETUP

We used an actor-critic setup for all RL experiments reported in this paper, following the distributed training and V-Trace algorithm implementations described by Espeholt et al. (2018). The distributed agent consisted of 1000 actors that produced trajectories of experience on CPU, and a single learner running on a Tensor Processing Unit (TPU), which learned a policy $\pi$ and a baseline $V$ using mini-batches of actors' experiences provided via a queue. The length of the actors' trajectories is set to the unroll length of the learner. Training was done using the RMSprop optimization algorithm. Please see the table below for values of fixed hyperparameters and intervals used for hyperparameter tuning.

| Hyperparameter | Values |
|---|---|
| Agent | |
|    Mini-batch size | [32, 128] |
|    Unroll length | [10, 40] |
|    Entropy cost | [1e−3, 1e−2] |
|    Discount $\gamma$ | [0.9, 0.95] |
| RMSprop | |
|    Learning rate | [1e−5, 4e−4] |
|    Epsilon $\epsilon$ | 1e−4 |
|    Momentum | 0 |
|    Decay | 0.99 |

Table 1: Hyperparameter values and tuning intervals used in RL experiments.

### A.3 ARCHITECTURE DETAILS

**Vision** – The visual system of our agents, which produced embeddings for state and goal from the raw pixel inputs, was identical to the one described in Espeholt et al. (2018), except ours was smaller. It comprised 3 residual-convolutional layers, each one with a single residual block, instead of two. The number of output channels in each layer was 16, 32 and 32. We used a smaller final linear layer with 64 units.

**Planner** – For the multi-head attention, queries, keys and values were produced with an embedding size of 64, using 1 to 4 attention heads. In our experiments, we did not observe a significant benefit from using more than a single attention head. The feedforward block of each attention step was a 2-layer MLP with 64 units per layer (shared row-wise). Both the self-attention block and the feedforward block were shared across iterations. After appending the state to the output of the final attention iteration, we used another MLP (shared row-wise) consisting of 2-layers with 64 units. The output of this MLP applied to each row was then aggregated using a max pooling, feature-wise, operation.

**Policy network** – The input to the policy network was the result of concatenating the output of the planner and the state-goal embedding pair (which can be seen as a skip connection). We used a 2-layer MLP with 64 units per layer followed by a ReLU and two separate linear layers to produced the policy logits ($\pi$) and the baseline ($V$).

### A.4 SCALABILITY

We experimented with two variants of the update function described in Section 4. In the first, all of the episodic memories attend to all of the others to produce a tensor with the same dimensions as the episodic memory. This variant, which we refer to as all-to-all (A2A), scales quadratically with the number of memories (and the number of steps per episode). The second version, which scales more favorably, takes the $k$ most recent memories from the episodic memory ($M$) as the initial belief state ($B$). On each self-attention iteration, each slot in $B$ attends to each slot in $M$, producing a tensor with the same dimensions as $B$ (i.e. $k$ vectors), which then self-attend to one another to produce the next belief state. The idea behind this design is that the querying from the $k$ vectors to the full memory can learn to select and summarize information to be composed via self-attention. The self-attention among the $k$ vectors may then learn to compute relationships, e.g. state-state paths, using this condensed representation. The benefit is that this architecture scales only linearly with the number of memories ($N$) in the episodic memory, because the quadratic time self-attention need only be applied over the $k$ vectors. This approach is similar to the inducing points of Lee et al. (2018a). Because this scales with $N * k$ rather than $N^2$, we call this variant N-by-k (abbreviated, Nxk).

Figure 5 compares the performance obtained on the Memory&Planning Game with the two architecture variants, A2A and Nxk. With $k$ set to one half of the original memory capacity ($k=50$), the Nxk agent recovers $92\%$ of the performance of the A2A. This result indicates that we can indeed overcome the quadratic complexity and opens up the possibility of applying EPNs to problems with longer timescales.

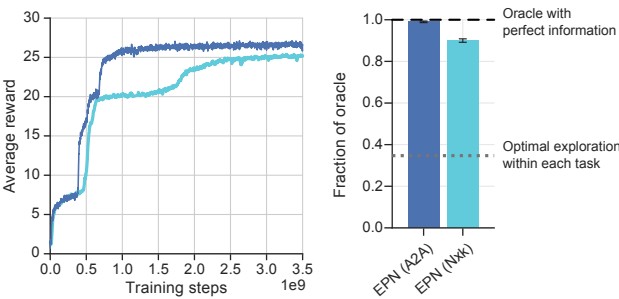

Figure 5: Comparison between architecture variants. The Nxk architecture variant, which scales linearly with the total number of memories, recovers 92% of the performance of the A2A variant, which scales quadratically.

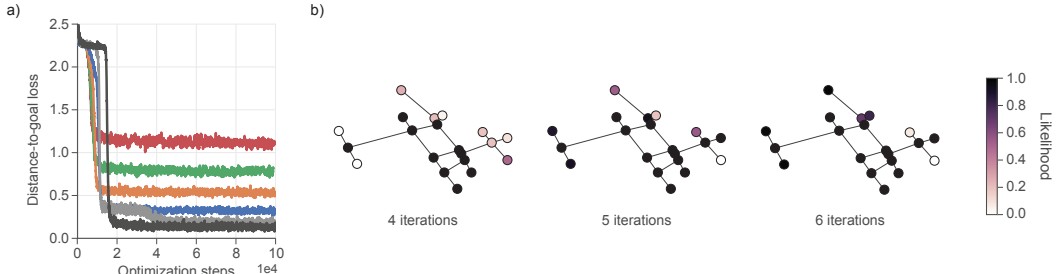

Figure 6: (a) Distance-to-goal decoding loss from the output of the planner (with frozen weights) after 1 to 6 iterations. Each additional iteration improves distance decoding. (b) The ability of EPNs to infer distance to goal expands with the number of self-attention iterations, even when that number goes beyond the number of iterations used during training of the EPN.

