# OpenReview forum: "Rapid Task-Solving in Novel Environments"
_ICLR.cc/2021/Conference — ICLR 2021 Poster_

### Official Review · AnonReviewer2 · 2020-10-28
**Marginal improvement on RTS illustrated on navigation without map**

**Rating:** 4
**Confidence:** 5

**Review:**

The paper is presenting a Transformer-based architecture for episodic memory-based sequential decision making.
The authors are interested in solving RTS and perform experiments that are focused on the case of mapless navigation.
The paper is insisting on the capacity to seamlessly coupling the exploration and planning parts of the overall task.
The authors are using a very pertinent dataset of google street maps for mapless navigation but seem to basically claim the superiority of Transformer against LSTM based policy in this particular setting which does not seem surprising.
A large quantity of work has been recently done on this task [1] [2] [3]
One remark is that the planning phase of the algorithm is completely embedded into the model which can be problematic depending on the nature of the task, especially regarding the possible need for decision explainability.
So, beyond parametric variation of tasks, as presented here, a good improvement could be to evaluate this approach on non-parametric variations of tasks as proposed in the manipulation framework of the Stanford meta world [4].

Refs
[1] Gated Path Planning Networks, 2018, Salakhutdinov and al
[2] LEARNING TO EXPLORE USING ACTIVE NEURAL SLAM, 2019, Salakhutdinov and al
[3] Semantic Curiosity for Active Visual Learning, 2020, Chaplot and al
[4] Meta-World: A Benchmark and Evaluation for Multi-Task and Meta Reinforcement Learning, 2019, Levine and al

---

> ### Author Response · Authors · 2020-11-12
> **Correcting a few misunderstandings**
>
> We thank the reviewer for their time in reviewing our paper. However, we want to be very clear that the review contains major misconceptions about the paper and its relation to the prior literature. Specifically,
>
> **RTS is not an established task.** The paper introduces and formalizes the problem for the first time. To the best of our knowledge, no previous work has approached the full RTS problem (including the work cited by the reviewer, see the following points) .
>
> **GPPN [1] does not explore.** These agents are given the full map when placed in a new environment. Additionally, GPPN's setting involves only one reward function per environment, so there is no notion of collecting information for future tasks, or repurposing previously acquired information for planning under a new reward function.
>
> **ANS [2], SC [3] do not plan.** ANS learns only to explore. In [2], the trained ANS module operates during an explicit exploration phase, then the agent agent is restarted in various locations and analytical planners navigate to target positions. This system would be inappropriate for the RTS setting where *there is no exploration phase*. In [3] the task does not require planning.
>
> **RTS is not meta-learning.** In RTS the reward function changes systematically within the inner meta-learning loop, requiring the agent to acquire and repurpose knowledge for planning *within a single episode*. The meta-learning benchmark [4] proposed by the reviewer does not apply to this setting.
>
> **Transformers vs LSTM is not the main comparison.** We show that EPNs outperform SOTA episodic memory-based agents, as well as LSTM-based agents. The former is by far the more insightful comparison. Additionally, the paper goes beyond showing that one architecture outperforms others – it investigates generalization capabilities, demonstrates the need for each architecture component, and analyzes the learned planning algorithms.
>
> We hope that this clears up the misconceptions underlying the initial review.

---

### Official Review · AnonReviewer4 · 2020-10-28
**Official Blind Review #4**

**Rating:** 7
**Confidence:** 3

**Review:**

### Summary

The authors present the challenge of rapid task-solving (RTS), where the goal is to solve a series of tasks as quickly as possible in a new, shared environment. This challenge requires both memory and planning capabilities from an agent, and the authors demonstrate that current SOTA modern deep RL agents with memory components still fail at this challenge. The authors instead propose “episodic planning networks” (EPNs), which contain architectural changes that allow for better long-term planning, and demonstrate their success on two challenges: a memory & planning game, and a navigation task from Street View images. Finally, the authors demonstrate that their proposed EPNs intuitively learn a planning algorithm similar to BFS, hinting at their generalization ability.

---

### Strengths

- The authors present a strong motivating example and an interesting challenge of rapid task-solving that seems both understudied and aspirational for current AI systems.
- The authors develop challenging but useful benchmark problems to compare their proposed method against, including the simpler planning game, and the more realistic and practical navigation task.
- Along with these benchmarks, the authors develop an interesting set of metrics, including the average training reward and the fraction of the oracle’s performance near the end of an episode, although the details of how these metrics were chosen and not others was less well-explained.
- The ablation experiment on how performance changes when replacing the iterative attention with a single pass is extremely thorough and well reasoned, along with showing strong empirical evidence for the authors’ claims.
- The authors clearly motivate why EPNs are necessary by demonstrating that current algorithms fail at the planning component, and that EPNs are able to optimally navigate and generalize to larger maps.
- The authors’ insight that EPNs recursively refine a reachability map, similar to value iteration, is an exciting insight into how compute can be scaled at test-time with the same training setting for better generalization.
- The qualitative visualization in Figure 4d of how the recursive refinement works is very convincing and easy to interpret.

---

### Weaknesses

- Some claims the authors make are only partially or weakly supported. For instance, the authors hypothesize that the two main reasons prior techniques fail is because the weighted sum of retrieved slots is not representationally expressive, and that the MLP is not computationally expressive to support planning. Are there ways to test these two hypotheses, ideally in isolation, potentially with ablative experiments?
- The description of the architecture in section 4 is hard to follow, although Figure 1b is very helpful. Can section 4 be formalized in terms of the specific modules and how they’re parameterized?
- I have some confusion around the metrics used to evaluate, specifically how these ones are chosen over other potential choices. For instance, the choice of 37% as the optimal exploration rate within each task, and the definition of oracle performance for the last third of an episode are not clearly explained.
- The authors note that the self attention network does not have access to the current state, only the goal state and episodic memories. This isn’t clear, as in diagram 1b it looks like the episodic storage has previous states as well as the current observed state, with the goal appended, and that the self-attention is happening across this entire vector. Some clarification around which keys and the features of those keys the self-attention is attending to would be helpful.
- Along the same note, the authors’ motivation for the self attention network not having access to the current state is that it would learn to compute a value map. How does performance change with a different design choice here, either with the goal state only available at the end and current state throughout, or have both the goal and current states available throughout the iterative self-attention?
- The authors also note that if the self-attention network did represent a reachability map, then a simple MLP on this map and the current state would be sufficient. Would a simple MLP on the map alone, not with the current state, also be sufficient? Specifically, if the map has information about all states, and the map states that the goal is not reachable from anywhere else in the map, would that not imply that the goal is reachable from the current location, even if the current location’s state is unknown?
- The authors claim that the specific self-attention function used was crucial. Ideally the authors would present ablations on this design choice, as the one they used looks similar to that used in transformers, with multi-headed attention and layernorm.
- The performance measure reported by the EPNs seems a bit suspect. Specifically, the baselines get to 5-10 tasks, but EPN gets to 28.7, where the median neighborhood has 12 locations. Does this mean that the EPN performance is with repeated starting positions or repeated goals? How novel are the latter set of tasks presented?
- The authors note in figure 4a that performance is roughly correlated with the number of iterations on self-attention. However, the authors do not explain why performance with 2 iterations converges to roughly the same final reward as with 4 iterations, and that 2 iterations converges more quickly. Further, it is a bit confusing that the performance is correlated with number of self-attention iterations for generalization (4b), and the distance to goal metric (4c), but that none of these are reflected in the average reward experiment (4a).
- A reference is made to figure 5a at the end of section 5 on the progressive removal of planning iterations -  is this meant to be figure 4a, or a different figure?
- In my mind, the comparison to Merlin, LSTM, and MRA serve as ablation studies. What would other baselines look like? Are there other techniques in meta-learning that could be applied here without an explicit architecture design of episodic memory or a planner? I would expect there to be several methods that could be applied to this setting - I don’t have a clear sense of what is SOTA in episodic model-based control, but would want to see those comparisons. As an example, “Explore then Execute: Adapting without Rewards via Factorized Meta-Reinforcement Learning” (DREAM) may have similar properties that make it a good comparison.

### Recommendation

Overall, I vote for accepting this paper. It demonstrates strong empirical results compared to previous SOTA in episodic model-based control. I do have some additional clarification questions on the experiments, and I wish the architecture section was more formalized. My main concern about this work is that it is hard to place in context with other potential baselines or areas of research, for example meta learning. Since I am not an expert on episodic model-based control, maybe this comparison is not realistic however.

---

> ### Author Response · Authors · 2020-11-17
> **Replies to Reviewer 4 (part 1/3)**
>
> Thank you for the thorough review and feedback. All of these are excellent points so we want to go through the weaknesses one by one.
>
> **R4:** "Some claims the authors make are only partially or weakly supported. For instance, the authors hypothesize that the two main reasons prior techniques fail is because the weighted sum of retrieved slots is not representationally expressive, and that the MLP is not computationally expressive to support planning. Are there ways to test these two hypotheses, ideally in isolation, potentially with ablative experiments?"
>
> Actually we have only a single hypothesis about the cause of MERLIN/RMA's failure in the RTS tasks: an MLP is insufficient _specifically_ when a weighted sum is used to aggregate memories. The EPN experiments test this hypothesis by replacing the weighted sum with a more sophisticated operation while leaving the MLP fixed. We take the fact that the resulting agent succeeds where the previous agents failed as evidence supporting the initial hypothesis. We will update the text to be clear that our hypothesis is specifically about _the combination of the weighted sum and MLP_. Thanks for bringing this lack of clarity to our attention.
>
> **R4:** "The description of the architecture in section 4 is hard to follow, although Figure 1b is very helpful. Can section 4 be formalized in terms of the specific modules and how they’re parameterized?"
>
> We will update the manuscript to improve the formalism and clarity of this section.
>
> **R4:** "I have some confusion around the metrics used to evaluate, specifically how these ones are chosen over other potential choices. For instance, the choice of 37% as the optimal exploration rate within each task, and the definition of oracle performance for the last third of an episode are not clearly explained."
>
> The optimal exploration performance level for the Memory&Planning Game resulted from a hand-coded policy that sweeps the map without repeating any visited state in the same task. This corresponds to the best an agent can do in an episode neglecting all the information acquired in previous tasks of that episode – it is the _optimal within-task_ performance. This performance level is particularly telling because it matches the performance achieved by the baseline (LSTM, RMA, MERLIN) agents.
> The oracle performance corresponds to a policy that always takes the shortest path from the current location to the goal location. This is only possible for an agent that is given privileged information about the full state-state connectivity of each new environment, which our trained agents do not have. To compare our agents’ performance to this perfect performance, and in particular, to measure their planning capabilities after some initial exploration, we chose the last third of the episodes, assuming that the initial two thirds of the episode would be partly required for the exploration.
> We will update the figures and methods sections to detail how optimal performance and oracle were computed, and the logic behind the oracle comparison.
>
> **R4:** "The authors note that the self attention network does not have access to the current state, only the goal state and episodic memories. This isn’t clear, as in diagram 1b it looks like the episodic storage has previous states as well as the current observed state, with the goal appended, and that the self-attention is happening across this entire vector. Some clarification around which keys and the features of those keys the self-attention is attending to would be helpful."
>
> This is a very good point that the diagram failed to communicate. In our experiments the episodic memory was only written to after the action for the current state was selected. This corresponds to a writing operation of **memory_t** that occurs at the beginning of **step_t+1**. We will update the diagram to reflect this.
>
> **R4:** "Along the same note, the authors’ motivation for the self attention network not having access to the current state is that it would learn to compute a value map. How does performance change with a different design choice here, either with the goal state only available at the end and current state throughout, or have both the goal and current states available throughout the iterative self-attention?"
>
> This is a good question and we don’t have a definitive answer to it. However the reason we made this particular choice was to prevent the SA from having the double duty of learning a good representation of the planning space and learning the planning itself. The two alternatives the reviewer proposes would allow this double duty to occur. Our solution, on the other hand, effectively separates these duties between the SA and the MLP, allowing for a more powerful (more parameters) and dedicated planning module, in the MLP.

---

> > ### Author Response · Authors · 2020-11-17
> > **Replies to Reviewer 4 (part 2/3)**
> >
> > **R4:** "The authors also note that if the self-attention network did represent a reachability map, then a simple MLP on this map and the current state would be sufficient. Would a simple MLP on the map alone, not with the current state, also be sufficient? Specifically, if the map has information about all states, and the map states that the goal is not reachable from anywhere else in the map, would that not imply that the goal is reachable from the current location, even if the current location’s state is unknown?"
> >
> > To be clear, with "reachability map" we mean something akin to a _value map_ that provides an estimate of the distance between any state and the goal. By design, this map does not contain information about which state the agent is currently in, even if the current state is one of the states in this value map. Without this information the policy network (MLP) would not be able to select actions to minimize distance, preventing the agent from planning effectively. We will update the paper to replace the phrase "reachability map" with "distance map" or similar, to prevent future confusion on this point.
> >
> > **R4:** "The authors claim that the specific self-attention function used was crucial. Ideally the authors would present ablations on this design choice, as the one they used looks similar to that used in transformers, with multi-headed attention and layernorm."
> >
> > To be more specific about the design decisions we were referring to here, the layer norm and residual connections were required and the order of these operations was important as well: the layer norm had to be applied to the inputs of each iteration, before applying the multi-head attention, followed by a residual connection. Applying the layer norm at the output of the multi-head attention, or removing the layer norm altogether, did not produce good results. We will update the paper to mention that these variants did not work well. It's worth noting that this ordering may have been critical because our attention system shares parameters across iterations, unlike typical transformers.
> >
> > **R4:** "The performance measure reported by the EPNs seems a bit suspect. Specifically, the baselines get to 5-10 tasks, but EPN gets to 28.7, where the median neighborhood has 12 locations. Does this mean that the EPN performance is with repeated starting positions or repeated goals? How novel are the latter set of tasks presented?"
> >
> > Starting and goal locations are independently sampled _with replacement_, as many times as needed until the episode step limit is reached. As a consequence, any agent will sometimes be tasked with reaching a goal state that was previously reached and this will happen more frequently for agents that perform better. To give you concrete numbers, in neighborhoods with 5 intersections (the training regime) there are ~20 states available to sample from. This results in 380 possible pairs of starting and goal locations and so it is unlikely that the same pair is sampled twice in any given episode. For neighborhoods with 12 intersections, this number jumps to ~2000. We are happy to include this information in the task description.

---

> > > ### Author Response · Authors · 2020-11-17
> > > **Replies to Reviewer 4 (part 3/3)**
> > >
> > > **R4:** "The authors note in figure 4a that performance is roughly correlated with the number of iterations on self-attention. However, the authors do not explain why performance with 2 iterations converges to roughly the same final reward as with 4 iterations, and that 2 iterations converges more quickly. Further, it is a bit confusing that the performance is correlated with number of self-attention iterations for generalization (4b), and the distance to goal metric (4c), but that none of these are reflected in the average reward experiment (4a)."
> > >
> > > This is an important observation that we should make clear in the paper. The reason for this is that performance for both 2 and 4 iterations is saturating. Because the agents are approaching optimal performance there is diminishing returns for increasing the number of iterations. To be clear, the performance indicated in the text (and figure 4a) is for problems involving neighborhoods with 5 intersections. If we increase the size of the neighborhoods the performance difference between 2 and 4 iterations becomes more pronounced. This effect is indicated by the divergence between the blue and green dashed lines in figure 4b. We will make the point about saturating performance in the 5 intersection regime more clear in the results section.
> > > For problems of the same size it is natural that self-attention with 2 iterations will converge more quickly than 4 because it corresponds to a model of the same capacity (number of parameters) but fewer layers to backprop through. We will make this point in the results section.
> > > We believe the confusion related to the performance correlation with the number of iterations stems from the fact that figure 4a is only showing performance on neighborhoods with 5 intersections, which was the training condition. This is in contrast to figures 4b and c, which show generalization performance on larger neighborhoods. We will add text to the Figure 4 caption to clarify this.
> > >
> > > **R4:** "A reference is made to figure 5a at the end of section 5 on the progressive removal of planning iterations - is this meant to be figure 4a, or a different figure?"
> > >
> > > Thank you for pointing this out. This is a typo that we will correct. We meant figure 4a.
> > >
> > > **R4:** "In my mind, the comparison to Merlin, LSTM, and MRA serve as ablation studies. What would other baselines look like? Are there other techniques in meta-learning that could be applied here without an explicit architecture design of episodic memory or a planner? I would expect there to be several methods that could be applied to this setting - I don’t have a clear sense of what is SOTA in episodic model-based control, but would want to see those comparisons. As an example, “Explore then Execute: Adapting without Rewards via Factorized Meta-Reinforcement Learning” (DREAM) may have similar properties that make it a good comparison."
> > >
> > > Although Merlin, MRA serve as ablations, as far as we know, they are also the current SOTA in tasks with the types of memory demands that we present here. Due to the long timescale over which memories must be stored and processed for planning, we focused on episodic memory architectures which had the potential to scale to very difficult memory problems.
> > > A key difference between our problem setting and the one described in DREAM is that ours does not have a separate exploration episode. DREAM relies on this clear distinction between exploration and execution phases in order to _separately optimize the exploration and execution policies_. In our work we were particularly interested in tasks that required a non-trivial mixture of exploration and goal-directed behavior, which we could optimize for with a single policy.

---

### Official Review · AnonReviewer3 · 2020-10-28
**Clear hypothesis and demonstration of the effectiveness of transformers as an episodic memory**

**Rating:** 7
**Confidence:** 4

**Review:**

The authors propose a non-parametric memory based on the transformer/self-attention architecture to learn over tasks that require planning from previously experienced tasks. They describe this style of learning as a form of meta reinforcement learning where individual episodes are a collection of tasks in the same environment and individual environments are sampled from a 'meta' environment. This is very similar in spirit to the RL-squared framework of Duan et al. 2017.

As is typical of meta-learning environments, there is an outer loop and an inner loop. The outer loop is composed of environments sampled from a 'meta' environment and within each sampled environment are tasks sampled with different initial states and goals within that environment.

Unlike the baselines in this paper, the proposed method utilizes a non-parametric memory that stores all (action, prev. state, observation) tuples that were experienced within an episode. This allows a model to leverage experiences from previous tasks to accomplish the current goal. The unique contribution presented in this work lies in using self-attention to integrate these past experiences.

Furthermore, results are demonstrated on a toy environment and a street view environment to demonstrate scalability to harder tasks.

The authors convincingly show that the proposed episodic memory architecture:

- Learns from experiences from past tasks by improving as time progresses within an episode
- Learning is improved when the self-attention memory is allowed more recurrent iterations
- Inner-loop learning is accomplished in the absence of gradient updates

Further improvements:

-Are multiple episodes present within each minibatch or are parameter updates computed from minibatches extracted from single episodes? Is this method sensitive to how the episodes/tasks are presented during training? In many settings, it is not be possible to have all environments available to sample from at any given training iteration due to limited computation or data resources.

-There is little to no discussion about the computational resources necessary as compared to the baseline methods described in this paper. The authors do provide some information in the supplementary, but these tradeoffs are likely significant enough to warrant some discussion in the main text. There is clearly a computational complexity difference between the self-attention mechanism and baseline models like the LSTM.

-How important is tuning the timescale of the environment in this work? How does the computational complexity and model performance scale with course and finer grains of timescale?

Duan, Yan, et al. "Rl $^ 2$: Fast reinforcement learning via slow reinforcement learning." arXiv preprint arXiv:1611.02779 (2016).

---

> ### Author Response · Authors · 2020-11-17
> **Replies to Reviewer 3**
>
> We thank Reviewer3 for the thoughtful feedback and positive review.
>
> **R3:** "Are multiple episodes present within each minibatch or are parameter updates computed from minibatches extracted from single episodes? "
>
>  We used an actor-learner set up (Impala) with multiple actors collecting experiences in parallel and asynchronously. For this reason, the majority of the trajectories in each mini-batch will come from different environments (different episodes).
>
> **R3:** "There is little to no discussion about the computational resources necessary as compared to the baseline methods described in this paper. The authors do provide some information in the supplementary, but these tradeoffs are likely significant enough to warrant some discussion in the main text. There is clearly a computational complexity difference between the self-attention mechanism and baseline models like the LSTM."
>
>  Thanks for this suggestion. We will use the extra space allowed in the camera ready to move the discussion of computational complexity into the main text. Additionally, we'll make the following point: Because the difficulty of the planning problems scales with the size of the environment, it's not realistic to hope that a constant time algorithm like the LSTM will be able to solve environments of arbitrary size. Indeed this is what we observed: even a 4x4 grid is too large for an LSTM-based agent. A key benefit of the EPN architecture is that it provides a way to scale the computation to the size of the problem – by scaling the number of memories and the number of iterations.
>
> **R3:** "How important is tuning the timescale of the environment in this work? How does the computational complexity and model performance scale with course and finer grains of timescale?"
>
> In environments with finer grained timescales, it would be appropriate to (1) store fewer memories by learning when to store, (2) learn to encode more than one timestep in a memory slot, and (3) use methods like the Nxk EPN that might learn to summarize multiple memory slots on-the-fly in service of planning. Using some or all of these methods, it may be possible to maintain full performance in environments with fine timescales, without excessively scaling the amount of compute required. An initial indication that this will be possible comes from the Nxk experiment, in which we observed that we could maintain close to full performance while drastically reducing the amount of computation and holding the timescale constant. We see extending the EPN approach to environments with finer timescales via the methods above as one of the most exciting directions for future research.

---

### Official Review · AnonReviewer1 · 2020-10-29
**Rapid Task-Solving in Novel Environments**

**Rating:** 8
**Confidence:** 4

**Review:**

The paper proposes 2 new benchmarks for Rapid Task Solving (RTS), that evaluate RL agents on the ability to memorize past experiences and learn to plan to solve new tasks in different environments rapidly. The paper also proposes Episodic Planning Networks (EPN), an RL method that replaces a weighted sum and multi-layer peceptrons in memory networks with self-attention.  The proposed EPNs proved to significantly outperform baseline methods with memory and an LSTM method without memory. The paper formulates RTS as an extension of meta-reinforcement learning framework, where in addition to optimizing over a distribution of tasks, the objective is to also optimize over a distribution over environments.

I think the work proposed in the paper is an important step towards developing general agents that learn to adapt quickly. To that end, it would be beneficial for the authors to release their code to benchmark new and older methods.

I have a few clarification questions.

How is the reward defined for each task in the 2 benchmarks? Is it 1 for reaching the goal and 0 otherwise?
Given that the 2 domains are defined in terms of relations between entities (connections between symbols in the Memory&Planning game and neighborhoods in the other) how would relational and symbolic RL methods perfom compared to the baselines and the proposed EPN? I believe these would provide a fairer baseline performance compared to the LSTM used.

---

> ### Author Response · Authors · 2020-11-17
> **Replies to Reviewer 1**
>
> We thank Reviewer1 for their useful feedback and positive review.
>
> **R1:** "it would be beneficial for the authors to release their code to benchmark new and older methods."
>
> Thanks for this suggestion. We will release code for sampling One-Shot StreetLearn tasks from the StreetLearn datasets, as well as a generator for the Memory&Planning Game.
>
> **R1:** "How is the reward defined for each task in the 2 benchmarks? Is it 1 for reaching the goal and 0 otherwise?"
>
> That's correct, in both benchmarks the agent receives a reward of 1 for reaching the goal and 0 otherwise. We will update the main paper text to include this information.
>
> **R1:** "how would relational and symbolic RL methods perfom compared to the baselines and the proposed EPN?"
>
> Relational RL: We wholeheartedly share the view that the planning problems we investigate are sensibly approached with *relational reasoning*; after all, the problem of shortest path finding can be cast as a problem of computing reachability relationships among states in a graph. As such, the classical field of relational RL [1] is relevant. However, most past work on relational RL has been tested only in the tabular setting and cannot be applied out-of-the-box to problems of interest in modern deep RL. A key exception is [2] which develops architectures that use weight sharing and self-attention mechanisms as inductive biases that encourage relational reasoning. In fact, our EPN architecture also uses weight sharing and self-attention for the same purpose, with the key difference that they operate over memories instead the observation, enabling the relationships among past states to be computed. Because it lacks this capability, the agent in [2] would not be suitable for our RTS tasks. To our knowledge, there aren't other relational RL agents that could serve as suitable baselines.
>
> Symbolic RL: As far as we know, symbolic methods for rl have so far only been shown to work in toy domains, specifically those that are fully-observed [3] and provide symbolic inputs [4]. As such, there isn't to our knowledge a symbolic method that would serve off-the-shelf as a suitable baseline. If such methods are developed in the future, we agree that it would be very interesting to see how they perform in the RTS setting.
>
> [1] Džeroski, S., De Raedt, L., & Driessens, K. (2001). Relational reinforcement learning. Machine learning, 43(1-2), 7-52.
>
> [2] Zambaldi, V., Raposo, D., Santoro, A., Bapst, V., Li, Y., Babuschkin, I., ... & Shanahan, M. (2018). Relational deep reinforcement learning. arXiv preprint arXiv:1806.01830.
>
> [3] Garnelo, M., Arulkumaran, K., & Shanahan, M. (2016). Towards deep symbolic reinforcement learning. arXiv preprint arXiv:1609.05518.
>
> [4] Dong, H., Mao, J., Lin, T., Wang, C., Li, L., & Zhou, D. (2019). Neural logic machines. arXiv preprint arXiv:1904.11694.

---

### Author Response · Authors · 2020-11-24
**Revision notes**

We thank all reviewers for the detailed reviews and helpful feedback. We have uploaded a revision to the paper which we hope will address most of your concerns.

We marked the changes to the text in red to make it easier to find them. In particular, we would like to call your attention to the revised version of section 4 which we partly rewrote following Reviewer 4's suggestion. We believe this section reads better now and has improved significantly in clarity. Thank you again for your feedback.

---

### Decision · Program_Chairs · 2021-01-07
**Final Decision**

**Decision:**

Accept (Poster)

**Comment:**

The paper proposes the challenge of rapid task-solving in unfamiliar environments and presents an approach to achieve this called Episodic Planning Networks -- a non-parametric memory based on the transformer architecture to learn tasks that require planning from previously experienced tasks, following a form of meta-RL.  The problem and approach are compelling, with strong empirical results.  The paper is well-written and is an exciting contribution.  This is a clear accept.

In response to the initial reviews, the authors updated their paper to improve the formalization and address other concerns in the reviews, which were viewed favorably by the reviewers as a good improvement. Based on the reviewer discussions, the work could still be placed better in context with respect to other literature.